# An Adaptive Multi-Target Jamming Waveform Design Based on Power Minimization

**DOI:** 10.3390/e22050508

**Published:** 2020-04-29

**Authors:** Jing Gao, Rihan Wu, Jinde Zhang

**Affiliations:** 1School of Control Engineering, Northeastern University at Qinhuangdao, Qinhuangdao 066004, China; 2School of Computer Science and Engineering, Northeastern University, Shenyang 110819, China; wurihan@stumail.neu.edu.cn (R.W.); 1871655@stu.neu.edu.cn (J.Z.)

**Keywords:** smart jammer, ground jammer, single-robust, double-robust

## Abstract

With increasing complexity of electronic warfare environments, smart jammers are beginning to play an important role. This study investigates a method of power minimization-based jamming waveform design in the presence of multiple targets, in which the performance of a radar system can be degraded according to the jammers’ different tasks. By establishing an optimization model, the power consumption of the designed jamming spectrum is minimized. The jamming spectrum with power control is constrained by a specified signal-to-interference-plus-noise ratio (SINR) or mutual information (MI) requirement. Considering that precise characterizations of the radar-transmitted spectrum are rare in practice, a single-robust jamming waveform design method is proposed. Furthermore, recognizing that the ground jammer is not integrated with the target, a double-robust jamming waveform design method is studied. Simulation results show that power minimization-based single-robust jamming spectra can maximize the power-saving performance of smart jammers in the local worst-case scenario. Moreover, double-robust jamming spectra can minimize the power consumption in the global worst-case scenario and provide useful guidance for the waveform design of ground jammers.

## 1. Introduction

Traditional jammers weaken the performance of enemy communication and radar systems by transmitting specific electromagnetic waves. With stepwise refinement of communication and radar systems, smart jammers have been gradually developed. Prior information ensures that smart jammers can adaptively transmit a waveform to prevent the radar system from effectively detecting targets. Presently, the transmitted waveform design of smart jammers is still under development; however, the radar waveform design method provides effective guidance for this.

As frequently used criteria for radar adaptive waveform design, signal-to-interference-plus-noise ratio (SINR) and mutual information (MI) are often employed as performance metrics for target detection and parameter estimation of radar systems. The estimation performance of the radar system can be optimized by maximizing the MI [1]. Matched illumination waveform design methods in signal-dependent interference based on signal-to-noise ratio (SNR) and MI for extended targets have been extensively studied [2,3]. Waveform design methods in signal-dependent interference have also been subsequently investigated; a joint receiving filter and waveform design method was proposed for a space-time adaptive processing (STAP) radar [4]. The system model of a distributed multiple-radar system (DMRS) coexisting with a wireless communication system was established in [5], and an optimal radar-transmitted waveform based on low probability of intercept (LPI) was designed. Additionally, radar waveform design methods in electronic warfare environments have been widely studied. Three different smart countermeasure models were considered; smart radar and dumb target, smart target and dumb radar, smart radar and smart target [6]. Since prior information of the target is rare in practice, the work in [7] assumed that the target spectra lie in an uncertainty set, limited by known upper and lower bounds; a waveform design method based on the minimum mean-square error (MMSE) and MI of multiple-input multiple-output (MIMO) radar was developed. Based on this, the angle-robust problem for co-located MIMO radar was addressed [8]. Overall, previous studies have laid a solid foundation for jamming waveform design methods.

### 1.1. Related Work

The research above provides effective guidance for adaptive jamming waveform design. For the MIMO radar, power allocation strategies of jamming waveforms based on MMSE and MI are addressed [9]. Additionally, channel-aware decision fusion (DF) in a wireless sensor network (WSN) is researched accounting for spectral efficiency [10]. The authors in [11] proposed orthogonal frequency division multiplexing (OFDM) radar jamming power allocation based on LPI performance for a joint radar and communication system. It is notable that LPI performance can be improved by minimizing power consumption. However, low power consumption may lead to interference performance degradation. Therefore, the jammer’s performance needs to be satisfied by setting constraints corresponding to the interference tasks before power management [12,13,14].

Several jamming waveform design algorithms have been proposed to improve the interference performance. In [15], two optimization algorithms of jamming waveforms for different jamming tasks were proposed; the detection performance and parameter estimation performance of the radar system were weakened by minimizing the SINR and MI, respectively. Furthermore, because precise characterization of the radar transmitted spectrum is rare in an actual electromagnetic environment, it is assumed that the radar-transmitted spectrum lies in an uncertainty set of spectra bounded by known upper and lower bounds, that significantly relaxes the prior knowledge of the radar-transmitted spectrum. Based on this fuzzy signal model, robust jamming techniques under both SINR and MI criteria have been researched extensively. However, integration of the ground jammer with the target was not realized. Precise characterizations of the radar-transmitted spectrum and target spectra are rare in practice for ground jammers. The robust jamming waveform proposed above limited the interference performance in the local worst-case scenario. The work in [16] proposed a robust joint design of the transmit waveform and filtering structure, where an iterative optimization procedure was advocated. The abovementioned studies primarily focused on optimization algorithms for a single target. However, radar systems or smart jammers need to detect multiple targets in practice. Therefore, waveform design based on multi-target optimization has been studied in recent years [17,18].

### 1.2. Major Contributions

In this study, we develop an adaptive multi-target jamming waveform design based on power minimization to improve a smart jammer’s power-saving performance. The key contributions of this study can be summarized as follows:(1)A power minimization-based jamming waveform design method for multiple targets is proposed. An optimal jamming waveform design is proposed based on the assumption that the jammer knows the exact knowledge of the radar-transmitted spectrum and target spectra. Then, considering that precise characterization of the radar-transmitted spectrum is rare in an actual electromagnetic environment, a single-robust jamming waveform design is investigated.(2)Furthermore, we recognized that the ground jammer is not integrated with the target. Therefore, precise characterizations of the radar-transmitted spectrum and target spectra are rare in practice for ground jammers. In order to meet the practical requirements of waveform optimization for the ground jammer, a double-robust jamming waveform design method is proposed in this paper.

The remainder of this paper is organized as follows. Section 2 introduces the considered system model and fuzzy signal model for multiple targets. The optimization models of optimal, single-robust, and double-robust jamming waveform design based on power minimization are established and solved in Section 3. Section 4 compares the presented algorithms according to simulation analysis. Finally, some conclusions are drawn in Section 5.

The notations used in this paper are as follows. The continuous time-domain signal is denoted by $w\left(t\right)$, the Fourier transform of $w\left(t\right)$ is $W\left(f\right)$, the symbol ∗ denotes the convolution operator, and different targets are denoted by i=1,2,3⋯.

## 2. System Model and Fuzzy Signal Model

As shown in Figure 1a, a coexistence scenario is considered, including radar, ground jammer, and multiple targets, in which the jammer is not integrated with the target. The radar transmits a specific waveform $w\left(t\right)$ to detect the target. Simultaneously, according to prior information acquired from the radar-transmitted spectrum and target spectra at the previous moment, the jamming waveform $j\left(t\right)$ is transmitted to disturb the radar and complete specific jamming tasks. The dotted lines of the target pointing to the radar and the jammer represent the target’s echo.

Figure 1b depicts the known target signal model. The target impulse response is denoted by $g_{i}\left(t\right)$, where i=1,2,3… denotes the different targets. The radar-transmitted waveform is represented by $w\left(t\right)$. Let $W\left(f\right)$ and $G_{i}\left(f\right)$ denote the Fourier transforms of $w\left(t\right)$ and $g\left(t\right)$ respectively. $J\left(f\right)$ is the power spectral density (PSD) of the jamming waveform $j\left(t\right)$, and $n\left(t\right)$ denotes a zero-mean channel noise process with PSD $S_{{}_{nn}}\left(f\right)$. The symbol ∗ denotes the convolution operator. The output corresponding to the target i is $y_{i}\left(t\right)$, which is denoted as
(1)$$y_{i}\left(t\right)=w\left(t\right)*g_{i}\left(t\right)+j(t)+n\left(t\right)$$

Owing to the spectrum estimation limitation in an actual environment, the spectra of the radar-transmitted waveform and target obtained by the smart jammer are fuzzy. To simulate the fuzziness of the radar-transmitted spectrum and target spectra, the band model presented in [19] is adopted. In this model, the upper and lower bounds of each frequency sample on the spectrum are represented by a margin of error, which is calculated by adding or subtracting a random number from the real value. The radar-transmitted spectrum is placed in an uncertainty class τ with known upper and lower bounds, which is denoted as
(2)$$\begin{array}{c} W\left(f\right)\in \tau \\ =\left\{l_{k}\leq \left|W\left(f_{k}\right)\right|\leq u_{k},for\;k=1,2,3\ldots K\right\} \end{array}$$
where the frequency samples are represented by $f_{k}$ and the band model is described in Figure 2. The solid blue line in the middle indicates the real radar spectrum.

Unlike the airborne jammer, the radar-transmitted spectrum and target spectra obtained by the ground jammer are all fuzzy. Furthermore, the radar system usually needs to detect multiple targets in an actual environment. Therefore, multiple targets are considered in this model. Similarly, the uncertainty class of multiple targets can be represented as follows [19]
(3)$$\begin{array}{l} G\left(f\right)\in \upsilon_{i}\\ =\left\{d_{ik}\leq \left|G\left(f_{k}\right)\right|\leq s_{ik},for\;k=1,2,3\ldots K\right\} \end{array}$$
where i=1,2,3… denotes the different targets. The uncertainty class $\upsilon_{i}$ for each target is different, and the number of targets is set to M. The occurrence probability of each target in this scenario is stochastic, and the corresponding target spectrum for each target is distinctive, but the sum of these occurrence probabilities is 1. It can be assumed that this scenario has four targets. The band model of [19] is shown in Figure 3. The solid lines in the middle indicate the real target spectra.

## 3. Power-Minimization Based Multi-Target Jamming Waveform Design

### 3.1. Optimal Jamming Waveform Design Based on Power Minimization

From the perspective of mathematical modeling, the optimal jamming waveform design based on power minimization is the problem of optimizing the jamming spectrum to minimize the power consumption subject to certain system constraints.

Optimal waveform design methods with two different system constraints are proposed. First, to effectively weaken the detection performance of the radar system, its output SINR should be limited by the threshold value. Intuitively, minimization of SINR means worse radar detection performance and higher power consumption. In an actual scenario, the SINR threshold can be adjusted according to the specific requirements of the jamming task. As indicated in [3], the output SINR of the radar system can be utilized as a metric for its target detection performance. The SINR can be expressed as
(4)$${\left(SINR\right)}_{t_{0}}=\int_{BW}\frac{{\left|W\left(f\right)\right|}^{2}\sigma_{g}^{2}\left(f\right)}{J\left(f\right)+S_{nn}\left(f\right)}df$$

Among them, $\sigma_{g}^{2}\left(f\right)$ denotes the spectra of multiple targets and can be expressed as [19].
(5)$$\sigma_{g}^{2}\left(f\right)=\sum_{i=1}^{M}p_{i}{\left|G_{i}\left(f\right)\right|}^{2}-{\left|\sum_{i=1}^{M}P_{i}G_{i}\left(f\right)\right|}^{2}$$

The spectrum of the target i can be expressed as $G_{i}\left(f\right)$, and $p_{i}$ denotes the probability of the occurrence of the target i [20]. For the radar system to be unable to complete the task of detecting targets normally, its detection performance should be limited. Mathematically, the output SINR of the radar system should be less than a specified threshold. Simultaneously, the power of the jamming spectrum should be minimized. Eventually, the optimization model of the optimal jamming spectrum design can be formulated as (6).
(6)$$\begin{array}{l} \min\int_{BW}J\left(f\right)df\\ s.t.\int_{BW}\frac{{\left|W\left(f\right)\right|}^{2}\sigma_{g}^{2}\left(f\right)}{J\left(f\right)+S_{nn}\left(f\right)}df\leq \gamma_{SINR} \end{array}$$

The parameter $\gamma_{SINR}$ represents the threshold value of the SINR. The jamming spectrum design strategy is convex, which can be solved analytically by employing the Lagrange multiplier technique. By introducing Lagrange multipliers λ>0 for this constraint, the objective function is denoted as
(7)$$K\left(\lambda ,J\left(f\right)\right)=\int_{BW}J\left(f\right)df-\lambda \left(\int_{BW}\frac{{\left|W\left(f\right)\right|}^{2}\sigma_{g}^{2}\left(f\right)}{J\left(f\right)+S_{nn}\left(f\right)}df-\gamma_{SINR}\right)$$

The formula above is equivalent to minimizing $K\left(J\left(f\right)\right)$, where $K\left(J\left(f\right)\right)$ is denoted as
(8)$$K\left(J\left(f\right)\right)=J\left(f\right)-\lambda \frac{{\left|W\left(f\right)\right|}^{2}\sigma_{g}^{2}\left(f\right)}{J\left(f\right)+S_{nn}\left(f\right)}$$

Because the second derivative of the function $K^{\prime \prime }\left(J\left(f\right)\right)$ with respect to the jamming spectrum $J\left(f\right)$ is less than 0, the expression of $J\left(f\right)$ can be generated by making the derivative test of the function $K\left(J\left(f\right)\right)$ equal to 0.
(9)$$J\left(f\right)=\max\left[0,D\left(f\right)\left(A-Q\left(f\right)\right)\right]$$
where $D\left(f\right)$ and $Q\left(f\right)$ are denoted as follows
(10)$$\left\{\begin{array}{l} D\left(f\right)=\left|W\left(f\right)\right|\sqrt{\sigma_{g}^{2}\left(f\right)}\\ Q\left(f\right)=\frac{S_{nn}\left(f\right)}{\left|W\left(f\right)\right|\sqrt{\sigma_{g}^{2}\left(f\right)}} \end{array}\right.$$
where A is a constant that can be defined by the SINR constraint.
(11)$$\int_{BW}\frac{{\left|W\left(f\right)\right|}^{2}\sqrt{\sigma_{g}^{2}\left(f\right)}}{\max\left[0,D\left(f\right)\left(A-Q\left(f\right)\right)\right]+S_{nn}\left(f\right)}df\leq \gamma_{SINR}$$

It is noted that both the radar-transmitted spectrum and the jamming spectrum are limited by bandwidth BW. The spectrum of the optimal jamming spectrum can be obtained by water injection on the function of $D\left(f\right)\left(A-Q\left(f\right)\right)$.

Similarly, the parameter estimation performance of the radar system can be limited by the MI constraint. Minimization of MI means worse radar parameter estimation performance and higher power consumption. The expression of MI between the received echo and the target impulse response can be expressed as [3]
(12)$$MI=Ty\int_{BW}\log\left(1+\frac{{\left|W\left(f\right)\right|}^{2}\sigma_{g}^{2}\left(f\right)}{Ty\left(J^{*}\left(f\right)+S_{nn}\left(f\right)\right)}\right)df$$
where Ty denotes the duration of the convolution output and the denotation of $\sigma_{g}^{2}\left(f\right)$ is expressed in (5). The optimization model of the MI constraint is established as follows
(13)$$\begin{array}{l} \min\int_{BW}J^{*}\left(f\right)df\\ s.t.Ty\int_{BW}\log\left(1+\frac{{\left|W\left(f\right)\right|}^{2}\sigma_{g}^{2}\left(f\right)}{Ty\left(J^{*}\left(f\right)+S_{nn}\left(f\right)\right)}\right)df\leq \gamma_{MI} \end{array}$$

The objective function can be expressed as follows
(14)$$K\left(\xi ,J^{*}\left(f\right)\right)=\int_{BW}J^{*}\left(f\right)df-\xi \left(Ty\int_{BW}\log\left(1+\frac{{\left|W\left(f\right)\right|}^{2}\sigma_{g}^{2}\left(f\right)}{Ty\left(J^{*}\left(f\right)+S_{nn}\left(f\right)\right)}\right)df-\gamma_{MI}\right)$$
where ξ>0 is the Lagrange multiplier for the constraint. Taking the derivative of $K\left(\xi ,J^{*}\left(f\right)\right)$ with respect to $J^{*}\left(f\right)$ and setting it to 0, the solution to be solved is
(15)$$J^{*}\left(f\right)=\max\left\{0,-Z\left(f\right)+\sqrt{Z^{2}\left(f\right)+R\left(f\right)\left[A^{*}-C\left(f\right)\right]}\right\}$$
where $Z\left(f\right)$, $R\left(f\right)$, and $C\left(f\right)$ are denoted as
(16)$$\left\{\begin{array}{l} Z\left(f\right)=S_{nn}\left(f\right)+\frac{\sigma_{g}^{2}\left(f\right){\left|W\left(f\right)\right|}^{2}}{Ty}\\ R\left(f\right)=\frac{\sigma_{g}^{2}\left(f\right){\left|W\left(f\right)\right|}^{2}}{Ty}\\ C\left(f\right)=\frac{S_{nn}{}^{2}\left(f\right)+\sigma_{g}^{2}\left(f\right){\left|W\left(f\right)\right|}^{2}S_{nn}\left(f\right)/Ty}{\sigma_{g}^{2}\left(f\right){\left|W\left(f\right)\right|}^{2}/Ty} \end{array}\right.$$

$A^{*}$ is a constant that can be defined by the MI threshold constraint.
(17)$$Ty\int_{BW}\log\left(1+\frac{{\left|H\left(f\right)\right|}^{2}{\left|X\left(f\right)\right|}^{2}}{Ty\left(J^{*}\left(f\right)+S_{nn}\left(f\right)\right)}\right)df\leq \gamma_{MI}$$

By using the first-order Taylor approximation for (15), a more intuitive solution can be obtained as follows
(18)$$J^{*}\left(f\right)=\max\left[0,G\left(f\right)\left(A^{*}-C\left(f\right)\right)\right]$$
where
(19)$$G\left(f\right)=\frac{{\left|H\left(f\right)\right|}^{2}{\left|X\left(f\right)\right|}^{2}/Ty}{2S_{nn}\left(f\right)+{\left|H\left(f\right)\right|}^{2}{\left|X\left(f\right)\right|}^{2}/Ty}$$

### 3.2. Single-Robust Jamming Waveform Design Based on Power Minimization

The above study assumes that precise characterizations of the radar-transmitted spectrum can be obtained. However, owing to the limitations of signal processing technology and the interference of clutter and noise in the actual environment, it is difficult for jammers to obtain accurate radar-transmitted spectra. The optimization model of a single-robust jamming spectrum based on a fuzzy signal model is proposed for the local worst-case scenario.

The upper bound of the uncertainty range of the radar-transmitted spectrum is $U\left(f\right)$, and the lower bound is $L\left(f\right)$. The SINR values in different cases are related as
(20)$$\int_{BW}\frac{{\left|L\left(f\right)\right|}^{2}\sigma_{g}^{2}\left(f\right)}{\overset{-}{J\left(f\right)}+S_{nn}\left(f\right)}df\leq \int_{BW}\frac{{\left|W\left(f\right)\right|}^{2}\sigma_{g}^{2}\left(f\right)}{\overset{-}{J\left(f\right)}+S_{nn}\left(f\right)}df\leq \int_{BW}\frac{{\left|U\left(f\right)\right|}^{2}\sigma_{g}^{2}\left(f\right)}{\overset{-}{J\left(f\right)}+S_{nn}\left(f\right)}df$$

When the upper bound of the radar-transmitted spectrum is selected, its output SINR is maximized. Therefore, the original optimization model in Equation (6) can be rewritten as
(21)$$\begin{array}{l} \min\int_{BW}\overset{-}{J\left(f\right)}df\\ s.t.\int_{BW}\frac{{\left|U\left(f\right)\right|}^{2}\sigma_{g}^{2}\left(f\right)}{\overset{-}{J\left(f\right)}+S_{nn}\left(f\right)}df\leq \gamma_{SINR} \end{array}$$

The proof method is consistent with the optimal solution under the SINR constraint. The single-robust jamming spectrum is denoted as
(22)$$\overset{-}{J\left(f\right)}=\max\left[0,\overset{-}{D\left(f\right)}\left(\overset{-}{A}-\overset{-}{Q\left(f\right)}\right)\right]$$
where $\overset{-}{D\left(f\right)}$ and $\overset{-}{Q\left(f\right)}$ are represented as follows
(23)$$\left\{\begin{array}{l} \overset{-}{D\left(f\right)}=\left|U\left(f\right)\right|\sqrt{\sigma_{g}^{2}\left(f\right)}\\ \overset{-}{Q\left(f\right)}=\frac{S_{nn}\left(f\right)}{\left|U\left(f\right)\right|\sqrt{\sigma_{g}^{2}\left(f\right)}} \end{array}\right.$$
where $\overset{-}{A}$ is a constant defined below
(24)$$\int_{BW}\frac{{\left|U\left(f\right)\right|}^{2}\sqrt{\sigma_{g}^{2}\left(f\right)}}{\max\left[0,\overset{-}{D\left(f\right)}\left(\overset{-}{A}-\overset{-}{Q\left(f\right)}\right)\right]+S_{nn}\left(f\right)}df\leq \gamma_{SINR}$$

The single-robust jamming spectrum can be obtained by water injection on the function of $\overset{-}{D\left(f\right)}(\overset{-}{A}-\overset{-}{Q\left(f\right)})$.

Similarly, the single-robust jamming spectrum with the MI constraint can be obtained according to the fuzzy signal model for the local worst-case scenario. The values of MI in different cases are related as
(25)$$Ty\int_{BW}\log\left(1+\frac{{\left|L\left(f\right)\right|}^{2}\sigma_{g}^{2}\left(f\right)}{Ty\left(\overset{-}{J^{*}\left(f\right)}+S_{nn}\left(f\right)\right)}\right)df\leq Ty\int_{BW}\log\left(1+\frac{{\left|W\left(f\right)\right|}^{2}\sigma_{g}^{2}\left(f\right)}{Ty\left(\overset{-}{J^{*}\left(f\right)}+S_{nn}\left(f\right)\right)}\right)df\leq Ty\int_{BW}\log\left(1+\frac{{\left|U\left(f\right)\right|}^{2}\sigma_{g}^{2}\left(f\right)}{Ty\left(\overset{-}{J^{*}\left(f\right)}+S_{nn}\left(f\right)\right)}\right)df$$

Therefore, when the upper bound of the spectrum of the radar-transmitted spectrum is selected, the MI value is maximized. The optimization model of a single-robust jamming spectrum with MI constraints is expressed as
(26)$$\begin{array}{l} \min\int_{BW}\overset{-}{J^{*}\left(f\right)}df\\ s.t.Ty\int_{BW}\log\left(1+\frac{{\left|U\left(f\right)\right|}^{2}\sigma_{g}^{2}\left(f\right)}{Ty\left(\overset{-}{J^{*}\left(f\right)}+S_{nn}\left(f\right)\right)}\right)df\leq \gamma_{MI} \end{array}$$

The proof method is consistent with the optimal solution under the MI constraint. The single-robust jamming spectrum is denoted as
(27)$$\overset{-}{J^{*}\left(f\right)}=\max\left\{0,-\overset{-}{Z\left(f\right)}+\sqrt{{\left|\overset{-}{Z\left(f\right)}\right|}^{2}+\overset{-}{R\left(f\right)}\left[\overset{-}{A^{*}}-\overset{-}{C\left(f\right)}\right]}\right\}$$
where $\overset{-}{Z\left(f\right)}$, $\overset{-}{R\left(f\right)}$, and $\overset{-}{C\left(f\right)}$ are denoted as
(28)$$\left\{\begin{array}{l} \overset{-}{Z\left(f\right)}=S_{nn}\left(f\right)+\frac{{\left|U\left(f\right)\right|}^{2}\sigma_{g}^{2}\left(f\right)}{Ty}\\ \overset{-}{R\left(f\right)}=\frac{{\left|U\left(f\right)\right|}^{2}\sigma_{g}^{2}\left(f\right)}{Ty}\\ \overset{-}{C\left(f\right)}=\frac{S_{nn}{}^{2}\left(f\right)+{\left|U\left(f\right)\right|}^{2}\sigma_{g}^{2}\left(f\right)S_{nn}\left(f\right)/Ty}{{\left|U\left(f\right)\right|}^{2}\sigma_{g}^{2}\left(f\right)/Ty} \end{array}\right.$$

$\overset{-}{A^{*}}$ is a constant that can be defined by the MI threshold constraint
(29)$$Ty\int_{BW}\log\left(1+\frac{{\left|U\left(f\right)\right|}^{2}\sigma_{g}^{2}\left(f\right)}{Ty\left(\overset{-}{J^{*}\left(f\right)}+S_{nn}\left(f\right)\right)}\right)df\leq \gamma_{MI}$$

According to the first-order Taylor approximation for (27), a more intuitive solution can be obtained as follows
(30)$$J^{*}\left(f\right)=\max\left[0,\overset{-}{G\left(f\right)}(\overset{-}{A^{*}}-\overset{-}{C\left(f\right)}\right]$$
where
(31)$$\overset{-}{G\left(f\right)}=\frac{{\left|U\left(f\right)\right|}^{2}\sigma_{g}^{2}\left(f\right)/Ty}{2S_{nn}\left(f\right)+{\left|U\left(f\right)\right|}^{2}\sigma_{g}^{2}\left(f\right)/Ty}$$

### 3.3. Double-Robust Jamming Waveform Design Based on Power Minimization

In the previous section, considering that the radar-transmitted spectrum exists in an uncertain class, the single-robust jamming spectrum is designed for the local worst-case. However, the above research assumes that the jammer can accurately obtain prior information of the target. The ground jammer is not integrated with the target; thus, the target spectra intercepted by the jammer are also fuzzy. Considering the above situation, an optimization model of a double-robust jamming spectrum is proposed in the global worst-case. By bounding this global worst-case performance, the overall performance will not be worse than this limit.

It is noted that the upper bound of the uncertainty range of multi-target spectra is $\sigma_{s}^{2}\left(f\right)$, and the lower bound is $\sigma_{d}^{2}\left(f\right)$. The SINR values in different cases are related as follows
(32)$$\int_{BW}\frac{{\left|L\left(f\right)\right|}^{2}\sigma_{d}^{2}\left(f\right)}{\overset{=}{J\left(f\right)}+S_{nn}\left(f\right)}df\leq \int_{BW}\frac{{\left|W\left(f\right)\right|}^{2}\sigma_{g}^{2}\left(f\right)}{\overset{=}{J\left(f\right)}+S_{nn}\left(f\right)}df\leq \int_{BW}\frac{{\left|U\left(f\right)\right|}^{2}\sigma_{s}^{2}\left(f\right)}{\overset{=}{J\left(f\right)}+S_{nn}\left(f\right)}df$$

Therefore, when the upper bounds of the radar-transmitted spectrum and the multi-target spectra are selected, the output SINR of the radar system is maximized. The upper bound of the multi-target spectra is denoted as
(33)$$\sigma_{s}^{2}\left(f\right)=\sum_{i=1}^{M}p_{i}{\left|S_{i}\left(f\right)\right|}^{2}-{\left|\sum_{i=1}^{M}P_{i}S_{i}\left(f\right)\right|}^{2}$$
where $S_{i}\left(f\right)$ denotes the upper bound of the target spectra. Therefore, the original optimization model in Equation (6) can be rewritten as
(34)$$\begin{array}{l} \min\int_{BW}\overset{=}{J\left(f\right)}df\\ s.t.\int_{BW}\frac{{\left|U\left(f\right)\right|}^{2}\sigma_{s}^{2}\left(f\right)}{\overset{=}{J\left(f\right)}+S_{nn}\left(f\right)}df\leq \gamma_{SINR} \end{array}$$

The proof is consistent with the optimal solution under the SINR constraint. The double-robust jamming spectrum is denoted as
(35)$$\overset{=}{J\left(f\right)}=\max\left[0,\overset{=}{D\left(f\right)}\left(\overset{=}{A}-\overset{=}{Q\left(f\right)}\right)\right]$$
where $\overset{=}{D\left(f\right)}$ and $\overset{=}{Q\left(f\right)}$ are denoted as follows
(36)$$\left\{\begin{array}{l} \overset{=}{D\left(f\right)}=\left|U\left(f\right)\right|\sqrt{\sigma_{s}^{2}\left(f\right)}\\ \overset{=}{Q\left(f\right)}=\frac{S_{nn}\left(f\right)}{\left|U\left(f\right)\right|\sqrt{\sigma_{s}^{2}\left(f\right)}} \end{array}\right.$$
where $\overset{=}{A}$ is a constant defined below
(37)$$\int_{BW}\frac{{\left|U\left(f\right)\right|}^{2}\sqrt{\sigma_{s}^{2}\left(f\right)}}{\max\left[0,\overset{-}{D\left(f\right)}\left(\overset{-}{A}-\overset{-}{Q\left(f\right)}\right)\right]+S_{nn}\left(f\right)}df\leq \gamma_{SINR}$$

The optimal jamming spectrum can be obtained by water injection on the function of $\overset{=}{D\left(f\right)}(\overset{=}{A}-\overset{=}{Q\left(f\right)})$.

Similarly, the double-robust jamming spectrum with the MI constraint can be obtained in the global worst-case. The values of MI in different cases are related as
(38)$$Ty\int_{BW}\log\left(1+\frac{{\left|L\left(f\right)\right|}^{2}\sigma_{d}^{2}\left(f\right)}{Ty\left(\overset{=}{J^{*}\left(f\right)}+S_{nn}\left(f\right)\right)}\right)df\leq Ty\int_{BW}\log\left(1+\frac{{\left|W\left(f\right)\right|}^{2}\sigma_{g}^{2}\left(f\right)}{Ty\left(\overset{=}{J^{*}\left(f\right)}+S_{nn}\left(f\right)\right)}\right)df\leq Ty\int_{BW}\log\left(1+\frac{{\left|U\left(f\right)\right|}^{2}\sigma_{s}^{2}\left(f\right)}{Ty\left(\overset{=}{J^{*}\left(f\right)}+S_{nn}\left(f\right)\right)}\right)df$$

Therefore, when the upper bounds of the spectra of the radar and the target are selected, the MI value is maximized. The optimization model of the double-robust jamming spectrum with MI constraints is expressed as
(39)$$\begin{array}{l} \min\int_{BW}\overset{=}{J^{*}\left(f\right)}df\\ s.t.Ty\int_{BW}\log\left(1+\frac{{\left|U\left(f\right)\right|}^{2}\sigma_{s}^{2}\left(f\right)}{Ty\left(\overset{=}{J^{*}\left(f\right)}+S_{nn}\left(f\right)\right)}\right)df\leq \gamma_{MI} \end{array}$$

The proof is consistent with the optimal solution under the MI constraint. The double-robust jamming spectrum is denoted as
(40)$$\overset{=}{J^{*}\left(f\right)}=\max\left\{0,-\overset{=}{Z\left(f\right)}+\sqrt{{\left|\overset{=}{Z\left(f\right)}\right|}^{2}+\overset{=}{R\left(f\right)}\left[\overset{=}{A^{*}}-\overset{=}{C\left(f\right)}\right]}\right\}$$
where $\overset{=}{Z\left(f\right)}$, $\overset{=}{R\left(f\right)}$, and $\overset{=}{C\left(f\right)}$ are denoted as
(41)$$\left\{\begin{array}{l} \overset{=}{Z\left(f\right)}=S_{nn}\left(f\right)+\frac{{\left|U\left(f\right)\right|}^{2}\sigma_{s}^{2}\left(f\right)}{Ty}\\ \overset{=}{R\left(f\right)}=\frac{{\left|U\left(f\right)\right|}^{2}\sigma_{s}^{2}\left(f\right)}{Ty}\\ \overset{=}{C\left(f\right)}=\frac{S_{nn}{}^{2}\left(f\right)+{\left|U\left(f\right)\right|}^{2}\sigma_{s}^{2}\left(f\right)S_{nn}\left(f\right)/Ty}{{\left|U\left(f\right)\right|}^{2}\sigma_{s}^{2}\left(f\right)/Ty} \end{array}\right.$$

$\overset{=}{A^{*}}$ is a constant that can be defined by the MI threshold constraint
(42)$$Ty\int_{BW}\log\left(1+\frac{{\left|U\left(f\right)\right|}^{2}\sigma_{s}^{2}\left(f\right)}{Ty\left(\overset{=}{J^{*}\left(f\right)}+S_{nn}\left(f\right)\right)}\right)df\leq \gamma_{MI}$$

According to the first-order Taylor approximation for (40), a more intuitive solution can be obtained as follows
(43)$$\overset{=}{J^{*}\left(f\right)}=\max\left[0,\overset{=}{G\left(f\right)}\left(\overset{=}{A^{*}}-\overset{=}{C\left(f\right)}\right)\right]$$
where
(44)$$\overset{=}{G\left(f\right)}=\frac{{\left|U\left(f\right)\right|}^{2}\sigma_{s}^{2}\left(f\right)/Ty}{2S_{nn}\left(f\right)+{\left|U\left(f\right)\right|}^{2}\sigma_{s}^{2}\left(f\right)/Ty}$$

## 4. Simulation and Results

In this section, simulation results are provided to confirm the validity and practicality of the optimal, single-robust, and double-robust jamming spectra. Figure 4 and Figure 5 define the scopes of the radar-transmitted spectrum and multi-target spectra, respectively, where the solid lines in the uncertainty sets represent the real radar-transmitted spectrum and multi-target spectra. The power of the real radar transmitted spectrum is more distributed near the frequency points of −0.2, 0, and 0.4. The number of real targets are set to 4, the main power is allocated near the normalized frequencies −0.3, −0.1, 0.1, and 0.3 for each target of the real multiple targets, and the corresponding occurrence probability $P_{i}\left(i=1,2,3,4\right)$ of each target is set to 0.1, 0.2, 0.2, and 0.5 respectively. The spectrum amplitude of the upper bound is the random value plus the spectrum amplitude of the real waveform spectrum. Similarly, the lower bound is the real spectrum amplitude minus the random value. Figure 6 shows the real radar transmitted spectrum and noise PSD. The noise PSD is set to be 1 w, and is assumed that the values of $\gamma_{SINR}$ and $\gamma_{MI}$ change from 1 to 5. Note that the spectrum amplitude of the radar-transmitted spectrum at frequency points of −0.2 and 0.4 are the same. The relevant parameters are listed in Table 1.

Figure 7, Figure 8 and Figure 9 show the optimal, single-robust, and double-robust jamming spectra when $\gamma_{SINR}$ is set to 1, 3, and 5, respectively. The power of the jamming spectra is concentrated at the frequency points of −0.2, 0, and 0.4; the spectrum amplitude at 0.4 is significantly higher than that at −0.2. Considering that the corresponding occurrence probability of the fourth target is 0.5, the jamming spectra based on SINR is dependent on both the radar-transmitted spectrum and the multi-target spectra. Therefore, the resulting jamming spectra are consistent with the analytical solutions derived in (9), (22), and (35). The trend of the optimal jamming spectrum is similar to the single-robust and the double-robust jamming spectra, caused by the same optimization model. With the increase in the SINR threshold, the amplitude of the jamming spectra decreases gradually, due to the relaxation of the jamming requirements.

Figure 10, Figure 11 and Figure 12 show the optimal, single-robust, and double-robust jamming spectra when $\gamma_{MI}$ is set to 1, 3, and 5, respectively. The power of the jamming spectra is mainly concentrated at the frequency points of −0.2, 0, and 0.4; the spectrum amplitude at 0.4 is also higher than that at −0.2. Therefore, the resulting jamming spectra are consistent with the analytical solutions derived in (15), (27), and (40). Similar to the jamming spectrum under the SINR constraint, the trends of the optimal, single-robust, and double-robust jamming spectra under the MI constraint are consistent.

Figure 13 shows the trends of jammer power consumption as SINR and MI thresholds increase. According to this, less jamming power is required to weaken the detection performance of the radar system as the specified SINR and MI thresholds increase. It is noted that the power consumption of the four types of jamming spectra are different. Owing to the lack of prior information, the linear frequency modulation (LFM) jamming spectrum has the largest power consumption. Compared with the LFM spectrum, the power consumption of the adaptive jamming spectra is significantly reduced. This reduction means effective power management and reduced probability of the jamming spectrum being intercepted by the radar system.

Figure 14 depicts the power consumption comparison employing different algorithms with specified SINR and MI thresholds. As expected, the optimal, single-robust, and double-robust jamming spectra outperform the LFM jamming spectrum because the former are optimized by an adaptive waveform design method. As shown in Figure 14a, the proposed optimal, single-robust, and double-robust jamming spectra design methods enable a decrease in the jamming power to 65%–90% of that obtained by the LFM signal under the SINR constraint. Similarly, in Figure 14b, the power consumption of the proposed optimal, single-robust, and double-robust jamming spectra reduce the jamming power to 66%–85% of the LFM signal under the specified MI threshold.

As shown in Figure 14, the best power-saving performance is achieved when the optimal jamming spectrum is employed, in which, the smart jammer transmits minimum power under a predetermined SINR and MI constraint. Considering that the upper bound of the uncertainty class of the radar-transmitted spectrum is employed, the single-robust jamming spectrum limits the local worst-case performance to an acceptable limit. Figure 14 validates that the power-saving performance of the single-robust jamming spectrum is worse than that of the optimal jamming spectrum. Because precise characterizations of the radar-transmitted spectrum and the target spectra are rare in practice for ground jammers, double-robust jamming spectra are designed. When such a spectrum is employed, the overall performance will not be worse than this limit. As expected, the power-saving performance of the double-robust jamming spectrum is worse than both the optimal and the single-robust jamming spectrum. We can derive that the actual power consumption of the jammer should be more than the optimal jamming power and less than the double-robust jamming power.

Overall, the optimal jamming spectrum under SINR or MI constraints can effectively reduce the power consumption of jammers. However, it is difficult to obtain the optimal jamming spectrum owing to the limited capacity of interception and spectrum estimation of the smart jammer. To make this algorithm more practical, single-robust and double-robust jamming spectra are applied to weaken the detection and parameter estimation performances of the radar system and simultaneously improve the power-saving performance of the jammer. A double-robust jamming spectrum can optimize the performance of jammers in the global worst-case, when the double-robust jamming spectrum is employed and the power-saving performance will not be worse than this bound. This algorithm provides effective guidance for the waveform design of ground jammers based on power-minimization in an actual scenario.

## 5. Conclusions

This study investigates an adaptive jamming waveform design based on power-minimization. As the number of targets detected by the radar is not unique, this study can be extended to a multi-target environment. By establishing the optimization model, the power consumption of the jamming spectrum is minimized and the SINR and MI are guaranteed to be less than a specific threshold value. The optimal jamming spectrum can minimize the power consumption of the jammer on the premise of specific detection performance. Furthermore, a single-robust spectrum is designed assuming that the radar-transmitted spectrum lies in an uncertainty class bounded by known upper and lower bounds. Additionally, because precise characterizations of the radar-transmitted spectrum and target spectra are rare in practice for ground jammers, a double-robust jamming spectrum is designed. The global worst-case performance is obtained when employing the double-robust jamming spectrum and the actual power-saving performance will not be worse than this limit. In future work, the results of this study may be applied to smart jamming in sensor networks [20]. Furthermore, the defense and low interception performances of the smart jammer can be improved through other performance metrics.

## Figures and Tables

**Figure 1 entropy-22-00508-f001:**
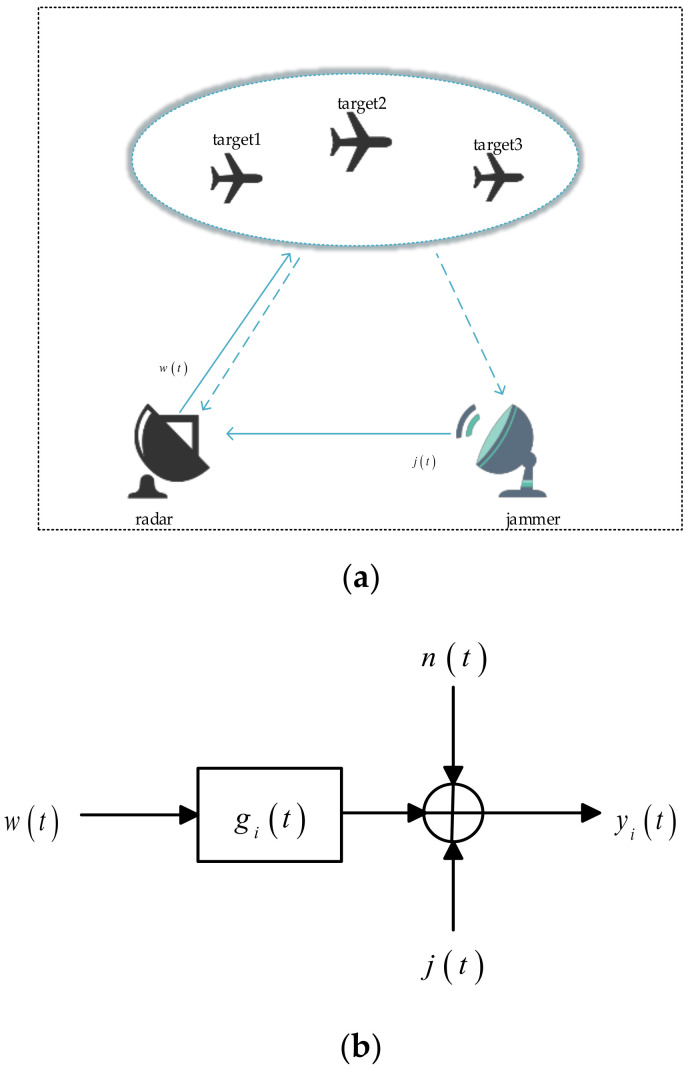
System and signal models. (**a**) System model with a ground jammer; (**b**) known target signal model.

**Figure 2 entropy-22-00508-f002:**
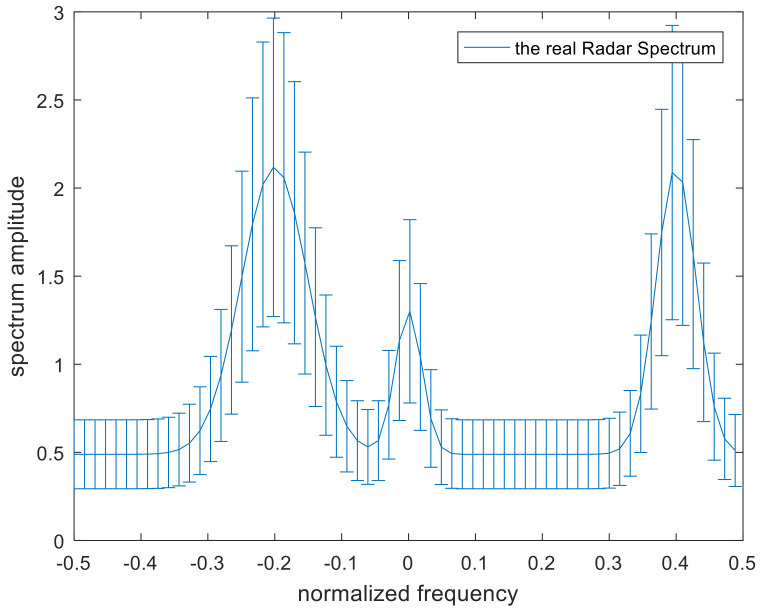
Band model of the fuzzy radar transmitted spectrum.

**Figure 3 entropy-22-00508-f003:**
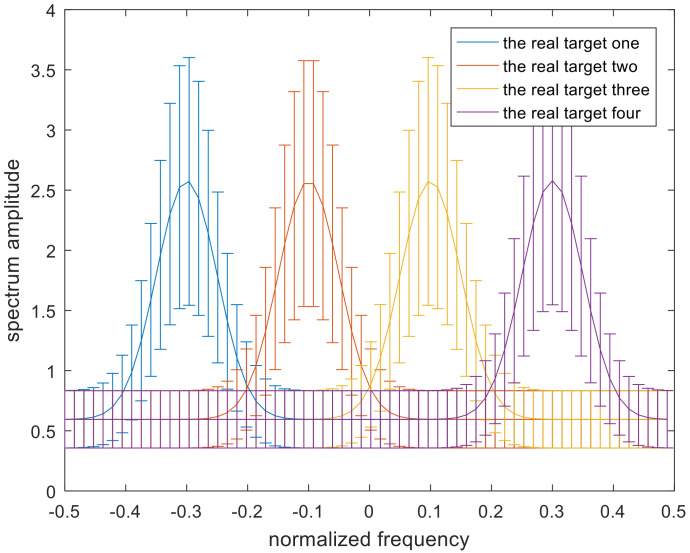
Model of the uncertainty range of the multiple targets.

**Figure 4 entropy-22-00508-f004:**
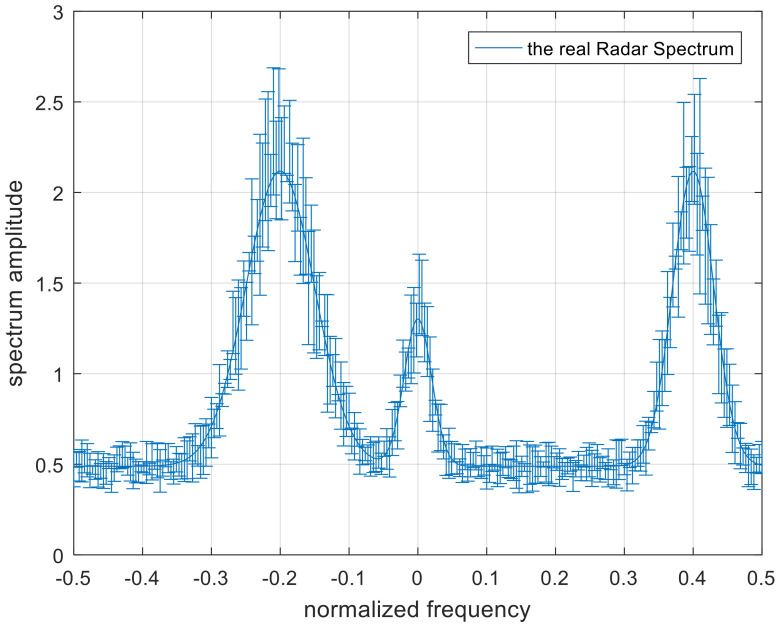
The uncertainty set of the radar spectrum.

**Figure 5 entropy-22-00508-f005:**
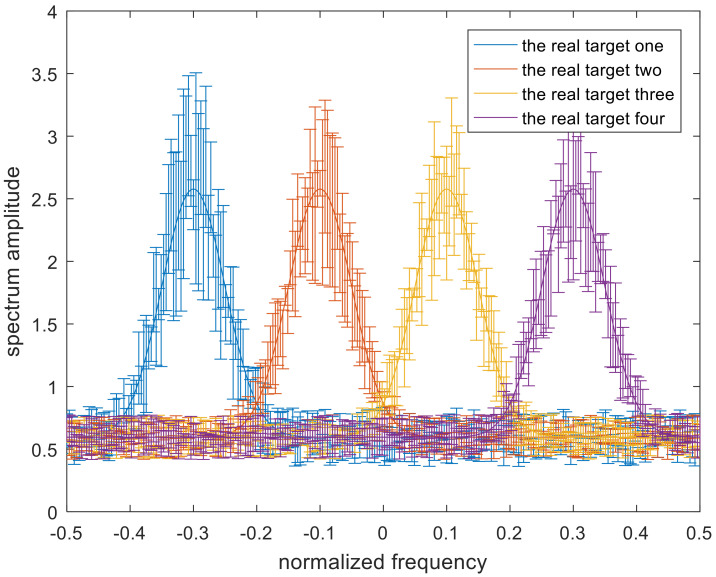
The uncertainty set of the target spectra.

**Figure 6 entropy-22-00508-f006:**
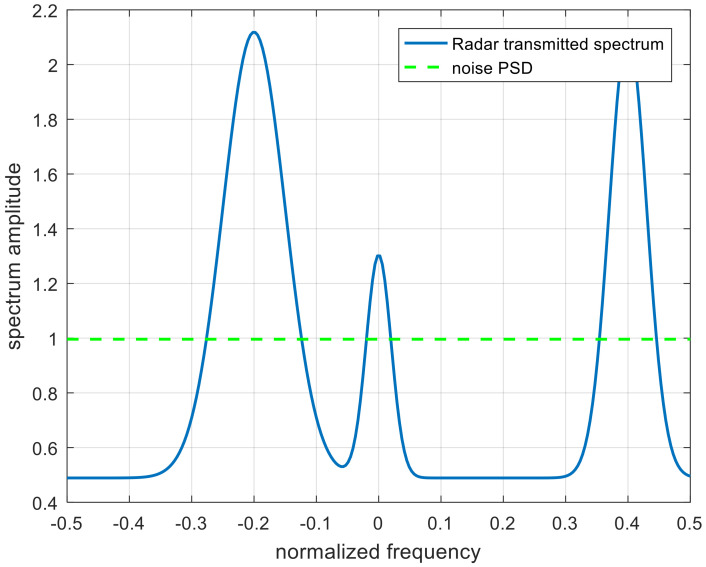
The spectra of real radar waveform and noise. PSD, power spectral density.

**Figure 7 entropy-22-00508-f007:**
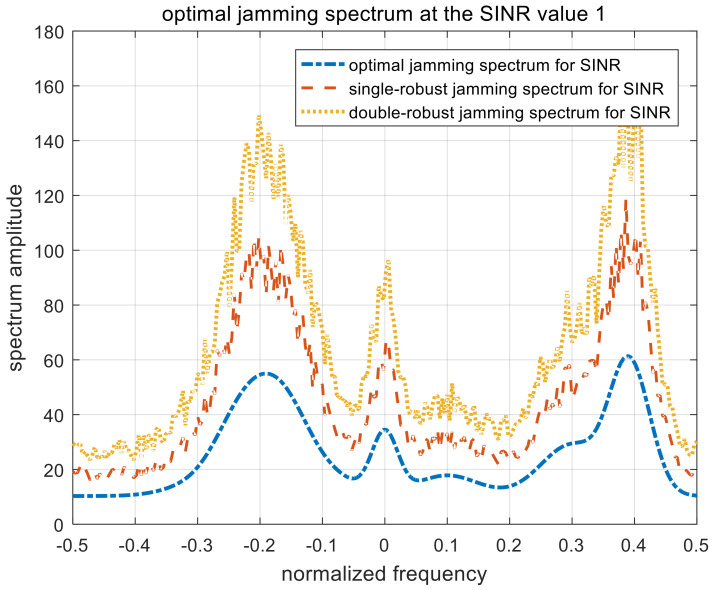
Optimal jamming spectra when $\gamma_{SINR}=1$. SINR, signal-to-interference-plus-noise ratio.

**Figure 8 entropy-22-00508-f008:**
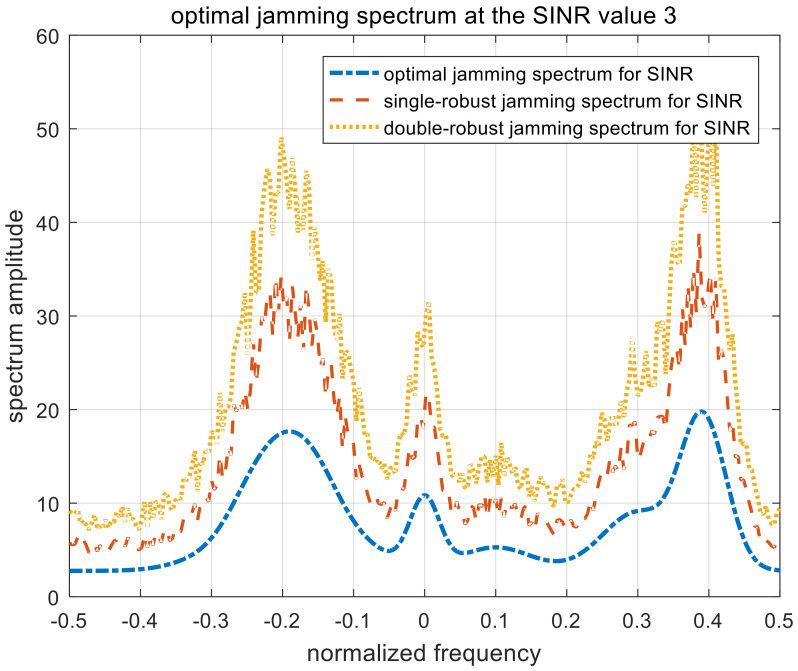
Optimal jamming spectra when $\gamma_{SINR}=3$.

**Figure 9 entropy-22-00508-f009:**
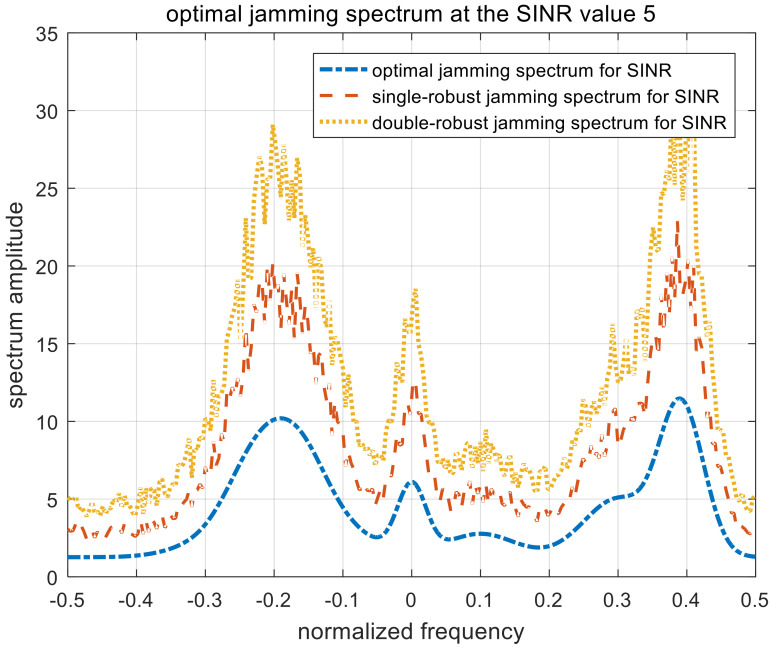
Optimal jamming spectra when $\gamma_{SINR}=5$.

**Figure 10 entropy-22-00508-f010:**
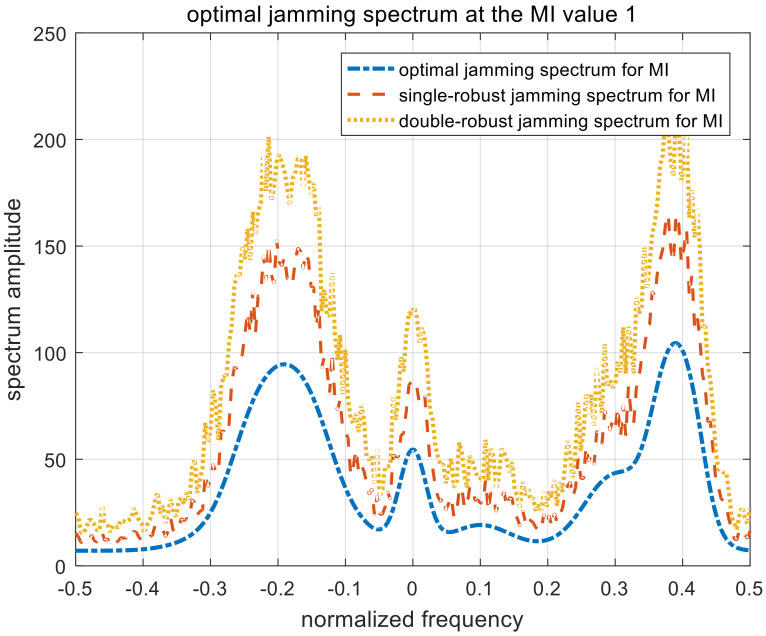
Optimal jamming spectra when $\gamma_{MI}=1$. MI, mutual information.

**Figure 11 entropy-22-00508-f011:**
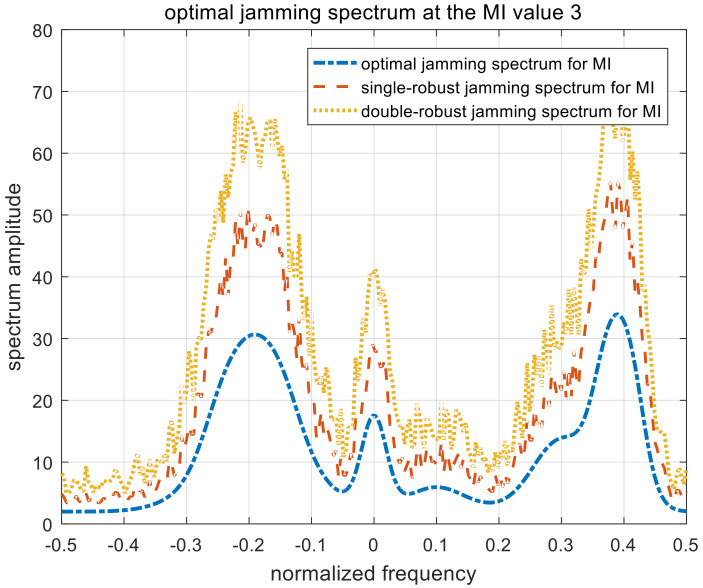
Optimal jamming spectra when $\gamma_{MI}=3$.

**Figure 12 entropy-22-00508-f012:**
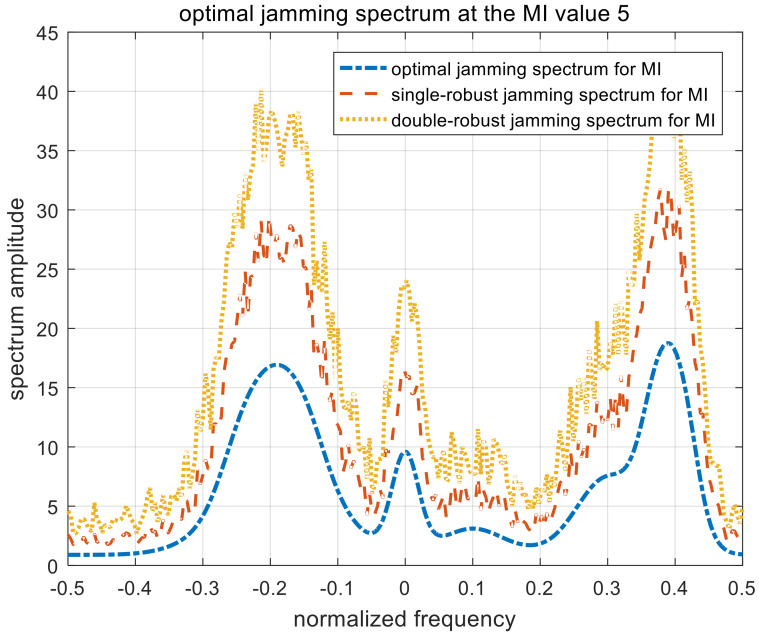
Optimal jamming spectra when $\gamma_{MI}=5$.

**Figure 13 entropy-22-00508-f013:**
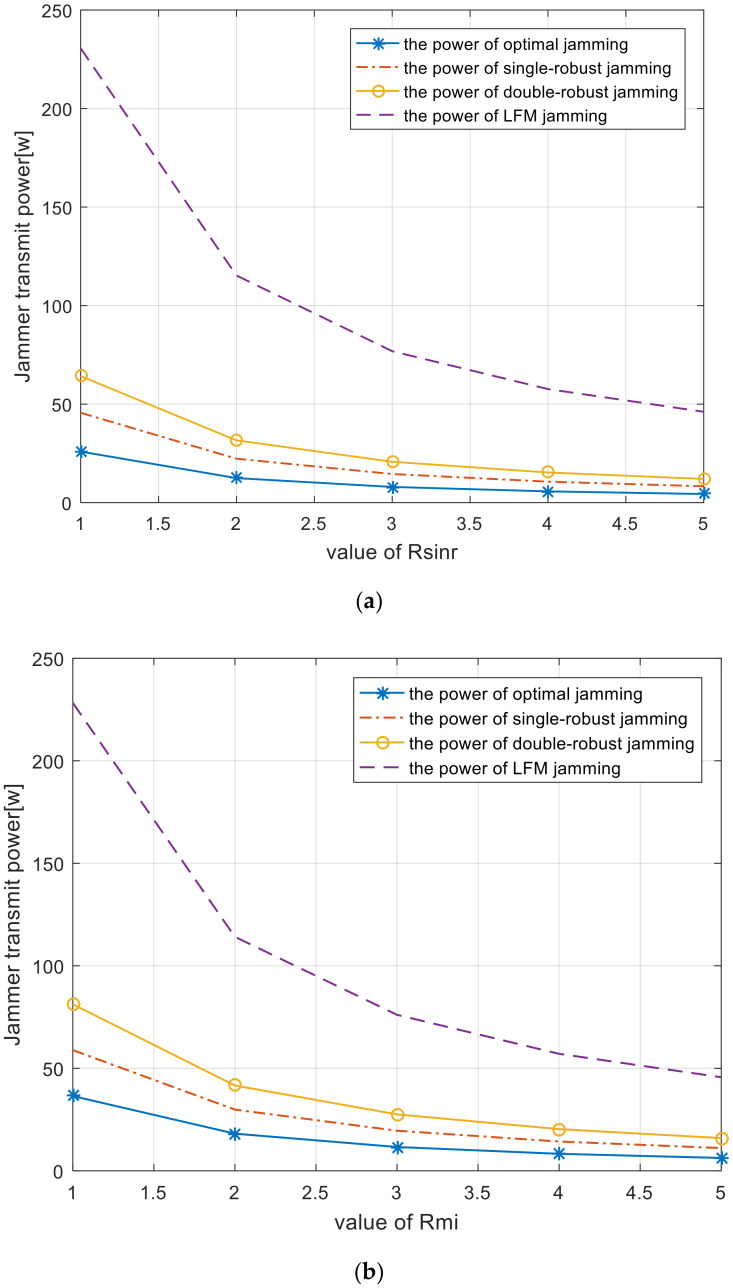
Total power of each jamming spectrum versus SINR and MI threshold (**a**) SINR; (**b**) MI.

**Figure 14 entropy-22-00508-f014:**
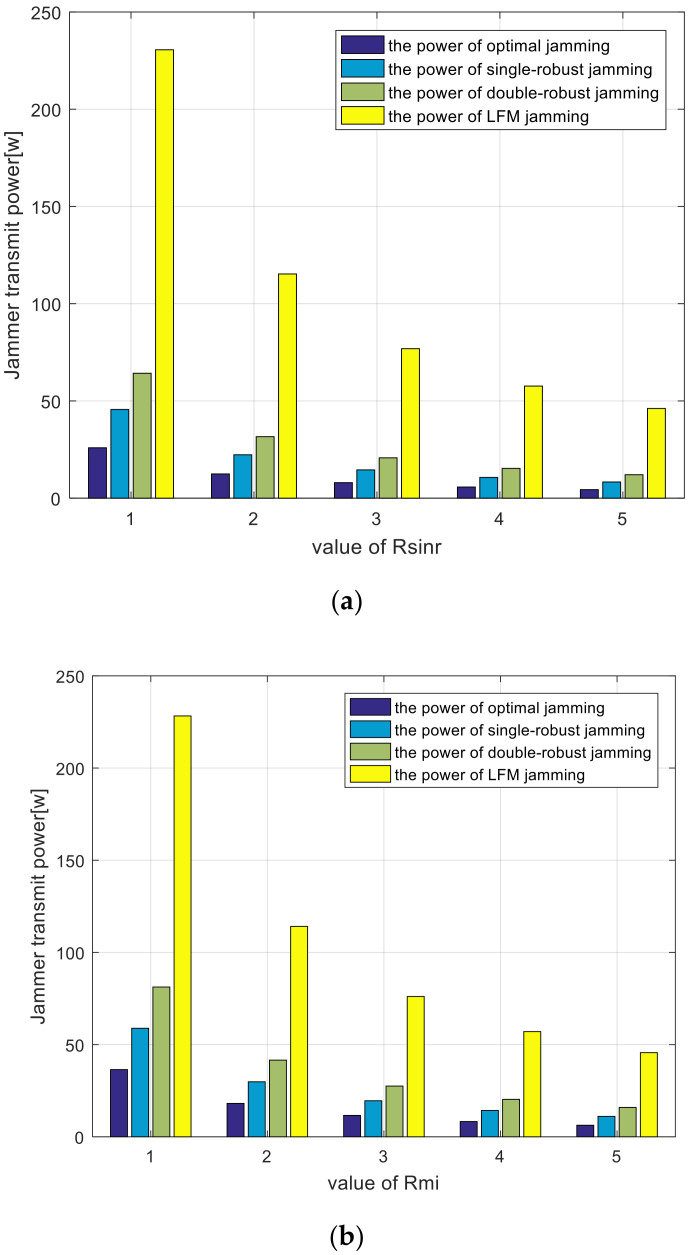
Comparisons of jamming power consumption employing different algorithms. (**a**) SINR constraint; (**b**) MI constraint.

**Table 1 entropy-22-00508-t001:** Relevant parameters.

Parameter	Value	Parameter	Value
$S_{nn}\left(f\right)$	1w	$P_{1}$	0.1
Ty	1s	$P_{2}$	0.2
$\gamma_{SINR}\;,\;\gamma_{MI}$	from 1 to 5	$P_{3}$	0.2
Δf	0.0039	$P_{4}$	0.5

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
