# Peer review of "An Adaptive Multi-Target Jamming Waveform Design Based on Power Minimization"

_entropy, 2020, doi:10.3390/e22050508_

Round 1
Reviewer 1 Report
Overall, this is an interesting paper on doubly-robust jammer waveform design in radar systems. However, before publication, it is my opinion that the following major comments should be addressed by the authors:
1) Abstract – “by a specified signal-to-interference-plus-noise ratio(SINR)” -> “by a specified signal-to-interference-plus-noise ratio (SINR)”.
2) Abstract – “low probability of intercept(LPI)” -> “low probability of intercept (LPI)”
3) Sec. I – “and ultra-low side-lobe antennas[6-7].” -> “and ultra-low side-lobe antennas [6-7].”
4) Please avoid the use of contracted forms in technical papers, e.g. “and target spectra can’t be captured accurately by the ground jammer” and the use of sloppy sentences, such as “Therefore, it is a good way to minimize the power while ensuring the detection task.”
5) Please add both notation and organization paragraphs at the end of Sec. I.
6) The statement of contributions in Sec. I should be rephrased and detailed so as to better highlight the technical challenges tackled by the authors (e.g. the use of fuzzy techniques). Similarly, the review of related works should be provided in a more streamlined and homogeneous fashion, e.g. avoiding the repeated use of “Author X has done Y”.
7) The following RWs on waveform optimization in the presence of uncertainties have been missed by the authors:
"Intrapulse radar-embedded communications via multiobjective optimization." IEEE Transactions on Aerospace and Electronic Systems 51.4 (2015): 2960-2974.
"Waveform design for radar STAP in signal dependent interference." IEEE Transactions on Signal Processing 64.1 (2015): 19-34.
"Subarray-based FDA radar to counteract deceptive ECM signals." EURASIP Journal on Advances in signal Processing 2016.1 (2016): 104.
"Robust waveform and filter bank design of polarimetric radar." IEEE Transactions on Aerospace and Electronic Systems 53.1 (2017): 370-384.
8) Sec. II – In my opinion, Fig. 1 should be redrawn so as to provide a graphical illustration of the whole considered system model. This would help the generic reader navigating through the description of the proposed approach.
9) The parameters describing the considered simulation setup should be summarized in a corresponding table. Additionally, the authors should improve the discussion part of Sec. IV, so as to make the main take-home message of each analysis spell out.
10) Similarly, I would also like the authors providing an assessment in the case of increased uncertainty in the radar spectrum.
11) The authors describe in Sec. I a number of related works on waveform optimization in the presence of jammers. However, none of them is employed as a baseline. At least a motivating discussion should be provided for such exclusion.
12) Conclusions should be enriched with what the authors consider to be further avenues of research. One possible direction may be the application of the proposed approach to smart-jamming in sensor networks, following:
"Rician MIMO channel-and jamming-aware decision fusion." IEEE Transactions on Signal Processing 65.15 (2017): 3866-3880.
"Jamming sensor networks: attack and defense strategies." IEEE network 20.3 (2006): 41-47.
Reviewer 2 Report
This paper entitled “An adaptive multi-target jamming waveform design based on power minimization” presents a study to optimize the spectrum of a jammer, based on power minimization, considering possible multiple targets. The paper is relatively well referenced, and the results give an idea of what we should expect in such a situation.
Nevertheless, the English needs to be reviewed throughout the text, as there are many typos that must be corrected, namely the verbs with plural not matching and things like “Three different smart countermeasures model…” on lines 34-35, or “Double-robust jamming waveform can optimizes…” on lines 270-271. So, all text must be revised also to better clarify the exposed ideas that sometimes are not very clear.
Some equations have variables or operators not defined. For example, Equation (4) with Snn(f), Equation (12) with Ty and Equation (14) with a Greek symbol ξ.
At line (117) is K”(J(f)) a second derivative?
The paragraph, between lines 146-148, seems to suggest that the ground jammer may be connected to the target. Would it be a condition for getting what you called a “double-robust jamming waveform”?
Throughout the text, as I just mentioned, and for example in lines 201-202, you always mention “jamming waveforms” when, in fact, you are talking about the signal’s spectra, that is, you are talking about signals in the frequency domain and not in the time domain, as the word “waveform” may suggest. Is there not a term to make it more clear?
Summarizing, I think your paper can be published, but there are a few things to polish first.
Round 2
Reviewer 1 Report
Overall, this is an interesting paper on doubly-robust jammer waveform design in radar systems. Additionally, the authors have carefully addressed all my previous comments and modified their manuscript accordingly.
Hence, I am glad to recommend the present work for publication within this journal.